# Evolution of Responses to COVID-19 and Epidemiological Characteristics in South Korea

**DOI:** 10.3390/ijerph19074056

**Published:** 2022-03-29

**Authors:** Junhwi Jeon, Changyong Han, Tobhin Kim, Sunmi Lee

**Affiliations:** Department of Applied Mathematics, Kyung Hee University, Yongin 17104, Korea; bm99093@khu.ac.kr (J.J.); cyhan@khu.ac.kr (C.H.); xhqls123@khu.ac.kr (T.K.)

**Keywords:** SARS-CoV-2, COVID-19, South Korea, infection-tree network, unlinked case, asymptomatic case, serial interval, effective reproduction number, social distancing, vaccination

## Abstract

The characteristics of COVID-19 have evolved at an accelerated rate over the last two years since the first SARS-CoV-2 case was discovered in December 2019. This evolution is due to the complex interplay among virus, humans, vaccines, and environments, which makes the elucidation of the clinical and epidemiological characteristics of COVID-19 essential to assess ongoing policy responses. In this study, we carry out an extensive retrospective analysis on infection clusters of COVID-19 in South Korea from January 2020 to September 2021 and uncover important clinical and social factors associated with age and regional patterns through the sophisticated large-scale epidemiological investigation using the data provided by the Korea Disease Control and Prevention Agency (KDCA). Epidemiological data of COVID-19 include daily confirmed cases, gender, age, city of residence, date of symptom onset, date of diagnosis, and route of infection. We divide the time span into six major periods based on the characteristics of COVID-19 according to various events such as the rise of new variants, vaccine rollout, change of social distancing levels, and other intervention measures. We explore key features of COVID-19 such as the relationship among unlinked, asymptomatic, and confirmed cases, serial intervals, infector–infectee interactions, and age/region-specific variations. Our results highlight the significant impact of temporal evolution of interventions implemented in South Korea on the characteristics of COVID-19 transmission, in particular, that of a high level of vaccination coverage in the senior-aged group on the dramatic reduction of confirmed cases.

## 1. Introduction

The emergence of the novel coronavirus (SARS-CoV-2) has posed the biggest epidemiological challenge to humans all around the world since the Spanish flu in the early 20th century. All countries were faced with implementing prompt prevention or mitigation strategies for the COVID-19 outbreaks [1]. It is difficult to design effective public policies because various complex factors (clinical/social/political) have become entangled with the novel COVID-19 virus, which continues to evolve, with new variants, including the recent delta [2] and omicron [3,4]. The fast evolution of viruses, in particular the omicron variant, makes it more difficult to keep the ongoing COVID-19 outbreaks under control, even with the timely development and allocation of vaccines in many countries [5,6]. Moreover, pre-symptomatic (infectious before symptom onset) or asymptomatic infections of COVID-19 make case isolation and contact tracing followed by quarantine less effective [7]. Combined with country-specific interventions, these hidden critical factors (usually unavailable or undetected when public health officials have to make critical decisions) have made profound effects on COVID-19 spread patterns and the distributions of morbidity/mortality in every country in a various manner [3].

As a result, COVID-19 has spread to 214 countries with 396,558,014 confirmed cases and 5,745,032 fatalities as of 9 February 2022 [8], such that the top 10 countries of most confirmed cases are in the United States of America, India, Brazil, France, United Kingdom, Russia, Turkey, Italy, Germany, and Spain [9], and the top 10 countries of most deaths are the USA, Brazil, India, Russia, Mexico, Peru, UK, Indonesia, Italy, and Iran, in the given order. Note that these rankings are subject to the population size and the testing coverage of each country. South Korea fares better with its rankings, with 57th and 75th of confirmed cases and deaths, respectively, due to the effective implementation of intensive interventions during the first and second waves [7]. The highest coverage of vaccination also plays a key role in managing the outbreak [10], even with the surge of delta variants since 18 April 2021. After relaxing social distancing measures on 1 November 2021, confirmed cases of COVID-19 have increased dramatically during the last two months in South Korea [10]. This partially coincides with the emergence of the new omicron variant and the decline of vaccine effectiveness [3,4,6].

There have been numerous research studies on clinical and epidemiological characteristics of COVID-19 over the last two years. The association of vaccination with symptomatic and asymptomatic SARS-CoV-2 infections was investigated among healthcare workers [11]. Assessment was carried out based on demographics, history of chronic illnesses, epidemiological/clinical characteristics, outcomes of linked and unlinked cases, and transmission potentials in different settings [12]. The impact of physical distancing interventions on incidence rate ratios of COVID-19 was explored using random effects meta-analysis synthesized across countries [13]. Detailed epidemiolocal characteristics were highlighted in Hong Kong for the first and second waves [14], and in Europe and Italy from January to December 2020 [15]. Age-specific confirmed cases of COVID-19 in Italy, South Korea, Spain, and the USA were compared [16,17], and age-dependent behaviors such as mask wearing, handwashing, physical distancing, crowd and restaurant avoidance, and cancellation of social activities were investigated [18].

In South Korea, based on extensive epidemiological investigation, the effective reproduction number, serial intervals, and age-specific transmission probability were compared for the first and second waves [19]. An infection network of COVID-19 was constructed, and the impacts of different interventions were explored during the first and second waves [20]. The study [21] analyzed demographics, transmission chains, case fatality rates, social activity levels and public health responses during the second and third waves in the Seoul metropolitan area. Since one of the common features of COVID-19 is cluster infection, the correlation between the types of clusters and the level of social distancing was investigated in [22]. Superspreading events were explored using a branching process model to measure the effective reproduction number and the dispersion parameter for the clusters in Seoul, South Korea from March to December 2020 [23]. These studies highlighted that there were significant superspreading events in the early and middle phases of COVID-19 outbreaks in South Korea.

Our study covers the longest period of COVID-19 infection in South Korea that has ever been conducted, as far as we know, from January 2020 to September 2021. The time span is divided into six periods to facilitate the observation of the evolution of infection characteristics such as the time-varying effective reproduction number, statistical information of unlinked and asymptomatic cases, infection networks, serial interval distributions, diagnostic delays and infection degrees, age-specific infector–infectee matrices, and infection profiles by age and region. We hope that our findings will provide helpful insights to formulate age/region/time-specific responses for future mitigation plans.

## 2. Materials and Methods

### 2.1. Data Sources

Epidemiological data were provided by the Korea Disease Control and Prevention Agency (KDCA) from 19 January 2020 to 16 September 2021 [24]. Epidemiological characteristics include overseas inflow, regional spread, infection route, date of confirmation, cluster classification, occupation, and relationship to prior confirmed cases. Clinical characteristics include symptomatic/asymptomatic state, date of symptom onset, etc. Weekly vaccination data from 26 February to 16 September 2021 include number of doses administered to various age groups, dosage information (1st, 2nd, and 3rd), and vaccine type (Pfizer, Janssen, Moderna, etc.) [24]. In addition, the number of tests, including the total number of daily tests, the number of new cases, and the test-positive rate, was provided by KDCA [25]. Lastly, the demographics and age statistics were obtained from the Ministry of the Interior and Safety of Korea (MOIS) and Korean Statistical Information Service (KOSIS) [26,27]. We partitioned the time span under our study of 20 months from 19 January 2020 to 16 September 2021 into six periods according to the characteristics of transmissions and major events in various interventions implemented by the Korean government, as listed in Appendix A.

### 2.2. Definitions

In this manuscript, confirmed cases (or just cases) are those with a positive result by real-time reverse transcription polymerase chain reaction (RT-PCR) tests. We excluded imported cases; thus, all cases that are analyzed in this study are local. A confirmed case is classified either as a linked case or an unlinked case, depending on the source of infection. The linked case is a local secondary case infected by a known confirmed case, while the unlinked case is a case whose source of infection is not identified or unknown [19]. Note the difference of unlinked cases and the unknown infection routes: if an individual is infected at a church gathering, for example, the case is classified as unlinked as long as the infector in the church is not identified. However, since it is certain that the infection occurred at the church, we can safely say that its infection route is the church.

The serial interval is defined as the time interval of symptom onsets of two successive linked cases [28,29]. The diagnostic delay is defined as the period from the symptom onset to the confirmation of infection. The degree of a case is defined as the number of secondary cases infected by the original case. The effective reproduction number $R_{t}$ is the average number of secondary cases generated by a primary case at time *t* [14,30].

### 2.3. Statistical Analysis

For the unlinked and asymptomatic cases, their weekly numbers were compared to those of the total cases to find their correlation through the Spearman coefficients. Similarly, the Pearson coefficients were computed for the daily numbers of aforementioned cases through linear regression. 

The time-dependent reproduction number $R_{t}$ measures an important epidemic indicator, which is the average number of secondary infected cases caused by a typical primary infected person at time *t*. In general, $R_{t}$ is estimated as the ratio of the number of new infections $I_{t}$ generated at time *t* to the infected individuals at time *t* in a renewal modeling approach given by $E\left[I_{t}\right]=R_{t}\overset{t}{\underset{s=1}{\sum }}I_{t-s}w_{s}$ [30]. We employed the Epyestim library in Python version 3.6.8 that implements the above process [31] using the serial interval as a gamma distribution (shape = 6.5, scale = 0.62), and a delay distribution as a discrete linear convolution of two one-dimensional sequences of the gamma distribution (shape = 1.35, scale = 3.77) and the negative binomial distribution (*n* = 0.63, *p* = 0.1). In addition, two smoothing options, LOWESS smoothing (21 days) and rolling average (3 days) were employed [31]. Although $R_{t}$ provides a good measurement of transmissibility of the disease, there are some difficulties such as the accurate specification of the generation interval and the reconstruction of the time series of new cases, which were discussed in detail in [32].

For the serial interval of each transmission pair, data that are between −15 days and 25 days were used to compute their statistics via the normal distribution fit using the maximum likelihood estimation [29], where the normality of the data was verified through the Shapiro–Wilk test. The population distributions by region and age were calculated by the average of the monthly population data. We compared the differences in the distribution of confirmed cases, unlinked cases, and asymptomatic cases obtained from the sources [17,26,27].

All analyses were performed in Python version 3.6.8.

## 3. Results

### 3.1. Progression of the Epidemic

We present the temporal trend of confirmed cases with their composition of unlinked and asymptomatic cases and the daily effective reproduction number $R_{t}$. Figure 1 shows the trend of weekly confirmed cases and the effective reproduction number $R_{t}$ where the entire study period is divided by five vertical dashed lines. Note that sustained rises of $R_{t}$ over 1 were observed to precede epidemic waves by a few weeks. We investigated the variation of $R_{t}$ in South Korea for a much longer period than [33] and [19], or in Hong Kong [14].

#### 3.1.1. Period 1 (P1): 19 January–29 April 2020

Since the first patient was reported on 19 January 2020, South Korea had a moderate increase in confirmed cases leading to weekly case numbers peaking at around 4200 in late February, with the majority of them found in the Daegu-Gyeongbuk area of southeastern Korea following a local church mass infection. About 85% of the confirmed cases could not identify their primary infection during the peak, but the unlinked cases sharply declined to around 35% as the spread was contained. After an initial drop around 18%, the asymptomatic case rate steadily rose as high as 57% until a sudden drop to 17%. Conversely, $R_{t}$ was high at the peak of the first wave but dropped dramatically to 0.3 in March. After a fluctuation of around 1 in April, $R_{t}$ skyrocketed over 3.5 with a wide range of uncertainty at the end of April. After the church incident, the number of daily confirmed cases gradually decreased, despite a couple of mass infections at another church and a call center in Seoul metropolitan area, to a single digit at the end of April.

#### 3.1.2. Period 2 (P2): 30 April–14 July 2020

South Korea maintained a low level of contagion, only interrupted by a series of small infections originating from some nightclubs in Seoul over the long weekend of 30 April–5 May. Although the nightclub incidents led up to sixth-order transmissions and other sporadic infections in the Seoul metropolitan area, the nationwide weekly case number was stable around 100–270. During this period, the ratio of unlinked cases and asymptomatic cases fluctuated in the range of 25–65% and 30–40%, respectively. $R_{t}$ also oscillated around 1 after plunging from the high value at the beginning of May.

#### 3.1.3. Period 3 (P3): 15 July–12 October 2020

The second epidemic wave started with the mass infection at a church (12% of total infection in P3) in Seoul, accounting for 12% of total confirmed cases of this period. The contagion was exacerbated by the major rally (additional 6.6% of total infection in P3) in Seoul on 15 August (8/15 rally). The government responded by upgrading the social distancing to level 2 (SDL2) in the Seoul metropolitan area the next day. As the spread worsened and was reported outside the metropolitan area, SDL2 was expanded nationwide on 23 August and was further raised to SDL2.5 in the metropolitan area on 30 August. The nationwide weekly case numbers peaked around 2100 in late August but eventually dropped to <400 after an intense application of quarantine program, and social distancing was eased to level 1 on 12 October. The asymptomatic case rate varied in the range of 25–45%, similar to that of P2, but during the second wave, the unlinked case rate and $R_{t}$ surged over 85% and 3, respectively.

#### 3.1.4. Period 4 (P4): 13 October 2020–25 February 2021

This period coincides with the third epidemic wave, and it started with a gradual increase in confirmed cases without any apparent major incidents. As the weekly case numbers were about to surpass 4000, the government upgraded SDL twice on 1 and 8 December as well as expanding the screening clinics to identify asymptomatic cases that were suspected to cause the third wave. Eventually, the weekly case number came to fluctuate around 3000 after peaking at around 7000 in late December. The asymptomatic case rate remained stable around 40% throughout this period, as did $R_{t}$ in the range of 0.7–1.5. On average, the unlinked case rate was the lowest moving in the range of 25–50%, but it notably hit the maximum in the same week as case numbers did.

#### 3.1.5. Period 5 (P5): 26 February–11 July 2021

South Korea commenced its vaccination on 26 February 2021 but also began to observe the new delta variant that was much more transmissible with an $R_{0}$ as high as 7 [4] versus the existing ones. Sporadic events of infection led to a gradual increase in confirmed cases and ultimately a major surge in early July. No major difference was observable for $R_{t}$ and the asymptomatic case rate, where the latter seemed to be slightly declining. Conversely, the unlinked case rate was steadily rising again as high as 80% in early July.

#### 3.1.6. Period 6 (P6): 12 July–16 September 2021

The country was hit by the fourth epidemic wave with 2000–4000 new infections registering in the daily figures, the largest to date. As in the third wave (in P4), it was no longer possible to pinpoint the exact sources of spread, while infections in everyday life accounted for the majority of confirmed cases. The more contagious delta variant was on the rise too. On 12 July, the social distancing was upgraded to level 4, which was the strictest policy thus far that allowed for gatherings of 4 or less before, and 2 or less after, 18:00.

Despite the overwhelming increase in confirmed cases, the asymptomatic case rate and $R_{t}$ were practically unchanged, hovering around 40% and 1, respectively. The unlinked case rate, however, remained high around 80%.

### 3.2. Asymptomatic and Unlinked Cases

Since the red curve in Figure 1a has roughly a similar shape to the profile of the bar graph, the unlinked case rate is suspected to have a positive correlation with the confirmed cases number. This is verified by the Spearman coefficients in Table 1 where we see significant positive correlations in each period with a sufficiently small *p* value (<0.05), i.e., except in P2 and P6. For the asymptomatic cases, only in P4 is the correlation reliably positive. It is either negative (in P1), or no reliable conclusion can be drawn (*p* > 0.05).

The regression lines in Figure 2 and Appendix A show that unlinked cases have stronger positive correlation to the confirmed cases than asymptomatic cases do, except for in P6. In each period, the Pearson correlation coefficients of unlinked cases are higher than those of asymptomatic cases, and the slopes of the regression lines also show the same property. For the unlinked cases in particular, the slope of the regression line and the Pearson coefficient are largest in P2 and P3, respectively. Note that in P1 and P2, we could observe the heteroskedasticity in both unlinked and asymptomatic cases with *p* values < 0.05 under the white test, and thus performed various treatments such as data logarithmizing, weighted regression, and RANSAC regression, without producing any definitive correlations. The cluster of outliers in P5 was due to our choice of the date that divides P5 and P6 according to the major change in the social distancing policy, which assigned some data with exploding numbers of cases in P5. Even if we had moved the cluster in P6, however, we were able to see the similar results in P5 and P6.

### 3.3. Infection Relationship

According to the relationship of each case with its infector, we partitioned the confirmed cases into five categories such as family, work, social, others, and unknown. If the data source discloses the relationship of a patient with the infector as family or “in the same household”, then the case is assigned to the family category. The work category includes source data with entries such as colleague, work, co-worker, business, staff, work-related, customer and so on. The infection in this category is estimated to happen in a typical workplace environment with commuting employees. We collected infected people whose infection contacts are acquaintance, friend, or relative, and assigned them to the social category, which shares similar contact profiles to private gatherings. The remaining types of interactions that occurred at various places, including “academy” (various types of cram schools for college admission, not a regular public/private school of education system) and nursing home, are grouped in the “others” category. Finally, the “unknown” category is for those whose place of infection was not disclosed at the time of reporting, i.e., they are the unlinked cases. Figure 3 shows a stacked bar graph of weekly confirmed cases (cf. Figure 1) segmented to represent these categories.

In view of Figure 3 and Table 2, the “unknown” category accounted for the highest percentage and had its peaks in P1, P3, and P6 when the first, second, and fourth waves occurred in that given order. The next largest category is social, except for in P2. While family and work occupy comparable proportions.

### 3.4. Serial Interval

Figure 4 shows the serial interval (the time interval of illness onsets between linked cases) distribution for each period. The serial intervals of less than −15 days or greater than 25 days are filtered out. The cases with negative serial intervals indicate the existence of pre-symptomatic infections, i.e., transmissions before symptom onset. The mean was between 2.91 (its lowest in P3) and 3.67 (its highest in P4), and the standard deviation was between 3.51 (its lowest in P6) and 5.83 (its highest in P1), where the statistics were obtained via the normal distribution fit after the normality test with *p* values of 0.245, 0.041, 0.378, 0.398, 0.422, 0.947 for each period in that given order when *n* = 25.

### 3.5. Diagnostic Delays and Degrees

The diagnostic delay of a case is its time span from symptom onset to positive (RT-PCR) test results, and the degree is the number of the secondary infections caused by a confirmed case. As shown in Figure 5 and Table 3, the delay shows a decreasing tendency with its median and mean attaining their maxima 6 and 6.58 in P1 and minima 3 and 3.51 in P6, respectively. Conversely, the degree oscillates between 0 and 1 for its median, and between 0.40 and 1.38 for its mean.

Table 4 shows no reliable correlation between the delays and the degrees in P1–4, but with the emergence of the delta variant, a positive correlation is shown in P4 and a negative correlation in P6.

### 3.6. Age Group and Infection

By decomposing the population into eight age groups such as “<20”, “20–29”, “30–39”, “40–49”, “50–59”, “60–69”, “70–79”, and “>79”, we analyzed the infections in each group in terms of various criteria. Figure 6 shows a stacked bar graph of weekly confirmed cases (cf. Figure 1) segmented to separate age groups, and we can see that as total cases surged in P6, each age group except for the older ones >69 did as well. 

More detailed information is available in Table 5. Cases of <20 exhibit a steady increase; those of 20–39 have their minima in P3 and increase afterward; 50–59 age group has the largest share of 20%, but this number remains stable throughout the entire period with a hint of drop in P6. Cases of >59 age groups significantly decline in P5–6 not only in their shares but also in absolute numbers. In terms of the shares out of the total cases, the youngest group <20 and the oldest group >79 have more linked cases than the others, while 20–39 age groups have more unlinked cases. In 40–79 the numbers are close. The confirmation delay ranges from 3.57–6.72 days for unlinked and 3.23–5.93 for linked cases. For the entire period, asymptomatic cases account for 31.5–39.4% of unlinked and 29.0–52.1% of linked cases. If a case is asymptomatic, it is more likely that its infector is identified than not in each period except for P1. Over time, the weekly proportion of women decreased in the linked cases (Spearman: −0.3403, *p* value: 0.0014), but it is difficult to say that there is a gender difference in the unlinked cases (Spearman: −0.1396, *p* value: 0.2082, excluding the first two weeks of no unlinked cases). 

When we try to understand the true nature of contagion according to age, the complete picture is only grasped after comparing the case numbers with the population size of age groups. Let $u_{i}=\left(u_{i,1},\;\cdots ,\;u_{i,8}\right)$ and $l_{i}=\left(l_{i,1},\;\cdots ,\;l_{i,8}\right)$ be the vectors of confirmed cases compartmented into age groups in the *i*th period that are unlinked and linked, respectively; let $p_{i}=\left(p_{i,1},\;\cdots ,\;p_{i,8}\right)$ be the vector of population of each age group in the same *i*th period. (That is, $u_{i,j}$, $l_{i,j}$, $p_{i,j}$ count the numbers in the *i*th period and *j*th age group, where we assigned indices j to age groups in the ascending order.) Then, the differences $\frac{u_{i}}{\sum_{j}u_{i,j}}-\frac{p_{i}}{\sum_{j}p_{i,j}}$ and $\frac{l_{i}}{\sum_{j}l_{i,j}}-\frac{p_{i}}{\sum_{j}p_{i,j}}$ of the infection distribution and the population distribution measures the relative prevalence of infection according to age groups [17]. Figure 7 shows these shares of confirmed cases relative to the populations size of age groups in each period. Minors (<20) were the most well-protected age group in P1 through P4, but had more linked cases than their share in P5 and P6. It is not apparent in Figure 6, but we can see in Figure 7 that the 20–29 age group was hardest hit in the first wave in P1, but fared well in the second and third wave in P3 and P4, only to succumb to the fourth wave in P6. People in age group 60–69 had the most infections in P2–4, but improved greatly in P6. The eldest group (>79) was most stable throughout the entire period.

Figure 8 shows the relative frequency of infector–infectee pairs with respect to their age for each period. The darker regions along the diagonals reveal that transmissions usually occurred among the people in the same age groups. Note that some of the maps are not symmetric with respect to the diagonals, indicating that infections can be directional. For instance, more people in the 40–49 infected people younger than 20 than vice versa in P1 and P4. It was the 50–59 age group that showed the most active infections in the same age group in P1, 3–5, while it was 60–69 in P2 and 20–29 in P6. Significant infections concentrated in the 40–69 age group in P1–5, while in P6, we can observe more activity in <30 than in 40–69.

### 3.7. Regions

South Korea administratively consists of 17 major divisions, including six metropolitan cities of Seoul, Busan, Incheon, Daegu, Daejeon, Gwangju, and Ulsan; large provinces of Gyeonggi, Gyeongnam, Gyeongbuk, Chungnam, Jeonnam, Jeonbuk, Chungbuk, and Gangwon; and small special districts of Jeju and Sejong. Seoul, Incheon, and Gyeonggi form the Seoul metropolitan area, accounting for half the total population (51.8 million as of 2020) of the country. As shown in Figure 9 and Appendix A, Seoul experienced the severest contagions in every period except for the first, and Gyeonggi was the second in P3–6. Note that Seoul is the most densely populated and Gyeonggi is the most populous region in South Korea (see Appendix A). Due to the Shincheonji church mass infection, Daegu and Gyeongbuk regions experienced the severest contagion in P1 but quickly had it under control in P2 as observed in [36]. We did not expect to see the pattern in P6 where the shares of unlinked and linked cases alternate with a large margin in most regions. Although Gyeonggi is the most populous region in South Korea, its population is distributed quite unevenly, with most people surrounding Seoul. We suspect that once we have a finer scale of population statistics such as the one using GIS-based location information [37], we will have a better understanding of spaciotemporal characteristics of transmission in South Korea.

## 4. Discussion

Numerous studies have recommended nonpharmaceutical interventions such as social distancing, school closure, remote working, and cancellation of social events to prevent a surge in cases and the overwhelming of healthcare facilities. To name a few, a proactive physical distancing [13] was found to be associated with a larger reduction of incidence rates, although the closure of public transportation had limited effect when other measures were in place. Combined isolation and tracing strategies [38] would reduce transmissions more than mass testing or self-isolation alone. A sustained implementation of strict social distancing, contact tracing, and household quarantine was required in order to safely reopen economic activities after a lockdown [39].

The first epidemic wave in South Korea started as the weekly case numbers first peaked at around 4200 in February 2020, with the majority of them found in the Daegu-Gyeongbuk area following a local church (Shincheonji) incident. Due to the inexperience of regional health authorities and the reclusive nature of the cultic church, however, they failed to execute proper contact tracing of the infected church members, and consequently an explosive spread of contagion swept through the Daegu-Gyeongbuk area of southeastern Korea. A series of minor infections in P2 originated from heavily populated nightclubs with poor ventilation that were prone to spread droplets carrying the virus. The young adults who visited the places were generally asymptomatic and were suspected to be “silent spreaders” of the outbreaks in their local communities. The second wave presumably originated from a church in Seoul (Sarang Jeil Church) where the church members’ singing in close contact during worship was the cause of spread. It was followed by the 8/15 rally whose attendants, mostly elderly, used public transit to spread the infection in local communities. These events support the claim [19] that the most frequent cluster types were religion-based activities. 

The third wave in December 2020 had no apparent trigger incidents. Some postulated the Halloween parties on 31 October or packed schools during several the weeks before the national entrance exam for university as possible epicenters, but no concrete link has ever been established. It is possible that the strengthened social distancing suppressed large mass-infection events but failed to control small and sporadic clusters [22]. Our finding that we had a higher share of unlinked cases in P3 than in P4 may sound contradictory to a study [21] that reported that compared to the second wave in P3, unknown routes of transmission were higher in the third wave that belongs to P4. However, we have to be careful to understand the difference of the two groups; while unlinked cases have no identifiable infectors, their transmission routes such as local clusters, imported, and hospitals can be established. Therefore, we may have way more unlinked cases than unknown routes of transmission in the event of mass outbreaks at certain facilities.

Limited supplies of the vaccine in South Korea necessitated formulation of optimal allocation strategies to mitigate morbidity and mortality at the same time. South Korea adopted the policy of maximizing first-dose administrations at the expense of timely application of second doses. This decision can find its support from a clinical study [40], which showed a four-fold reduction in asymptomatic infection in particular.

Figure 2 shows the correlation of case numbers and the rates of unlinked and asymptomatic cases, where we have found a positive correlation with unlinked cases but not with asymptomatic cases. Our results support previous studies such as [12] in Hong Kong that estimated the transmission potential of unlinked cases and [41] in Japan that concluded that the transmissibility of asymptomatic cases was limited. Our Spearman coefficients (0.78, 0.76, 0.51, and 0.56 in each period with *p* < 0.05) of unlinked cases were significantly higher than 0.39 of [12], indicating stronger positive correlation with total case numbers. It takes longer time for an unlinked case to be tested and quarantined than its linked counterpart; thus, the former has more opportunities to spread the virus and infect people. Conversely, asymptomatic cases were found [41] to be 3.7 times less infectious than symptomatic cases, which can negatively affect the number of total infections. Although this was observed in P1, we also have a positive correlation in P4 and no reliable results in the other four periods. Therefore, it is difficult to draw any definitive conclusion regarding the correlation of asymptomatic cases and the size of total cases. Note that P6 is the only time interval where we observe negative correlation between unlinked cases and total cases (though *p* > 0.05), which may be explained that as the contagion spread at an alarming rate, the government vigorously applied a test–trace–quarantine program, which ultimately outpaced the transmission. (cf. Appendix A) As reported in Israel [11] and the USA [42] that commenced vaccination earlier than South Korea, we also observe a significant reduction in incidence rates in the older populations whose vaccination started in P5.

If we exclude “unknown” cases whose infection routes were not identified, social gatherings were consistently the most common source of infection, followed by family and work. Institutions with large occupation capacity such as nursing homes and academies were not of significant concern. The shares of family and work categories were also more or less stable even during major outbreaks. We suspect that the share of social settings is highly variable since this category is most susceptible to the change of social distancing measures. Our results contrast dramatically with the studies [14,43] in Hong Kong where contagion within family members was the most serious by a large margin, followed by social (20.3% or 33.1%) and work (8.1% or 11.8%) environments. In the presence of unlinked cases that are the majority by a large margin, however, it is difficult to directly compare two regions’ results. In view of Figure 3 and Appendix A, the percentage of the unknown category appears to be inversely correlated with the number of tests per case in P1–3, although this tendency is reversed in later periods.

We carried out serial interval computations for a longer period than [19] and obtained a similar result that its mean is about 3 days (to be exact, 3.3, 3.2, 2.9, 3.7, 3.4, and 3.2 days in each period). While the contagions in P1–3 can be characterized by mass infections originating at particular settings such as a church in Daegu-Gyeongbuk area in P1, *n*^th^-order transmissions originating at nightclubs in Seoul in P2, and another church and 8/15 rally in Seoul in P3, the majority of those from P4 and onward can be classified as spreads in community level. Although the gradual decrease in serial intervals after peaking in P3 was concurrent with the heightened level of social distancing, we are hesitant to draw any causal conclusion due to the large volume of unlinked cases that do not allow for the extraction of serial interval information.

Note that our mean serial interval is shorter than 4.7 days [44] of the six regions combined (China, Germany, Hong Kong, Singapore, South Korea, Taiwan, and Vietnam) from December 2019 to February 2020, 4.74 days [14] of Hong Kong from January to August 2020, and 5.8 days [43] of Hong Kong from January to April 2020, which means that new cases were generated faster than before, and thus without an accelerated execution of trace–test–quarantine, we would face an imminent healthcare capacity overrun. 

We found that the diagnostic delay decreased from 6.58 to 3.51 as the pandemic spread throughout South Korea from January 2020 to September 2021 in comparison to 5.18 [14] in Hong Kong from January to August 2020. The delay was the longest in P1 (see Table 5) because in the early stages, the quarantine program was not properly prepared, and RT-PCR test kits were not widely available. As the logistics of personal protection equipment and the response system improved, however, we observe that the tails of delay in the heatmaps (cf. Figure 5) became lighter and lighter. Although the delay and the degree are in general positively correlated since an extended exposure of a patient to the public tends to generate larger secondary infections, we do not observe this trend in the heatmaps, which is also confirmed by Table 4. We suspect that the positive correlation in P4 is due to the emergence of the highly contagious delta variant, and the subsequent negative correlation in P6 is the result of the strongest quarantine policy as an administrative response and the strict observance of a quarantine program after onsets of symptom and subsequent tests. Similar results [43] in Hong Kong were reported such that shorter delays did not necessarily correlate with small local cluster sizes.

The infection degrees in South Korea were found to follow an extremely positively skewed distribution as in a previous study [20]. The low values of the degree can be attributed to the rapid execution of test–trace–quarantine program when contact tracing is in effect. Nonetheless, we still observed a sustained increase in confirmed cases because far more cases slipped away from contact tracing.

Unlike past pandemics that had higher mortality but lower morbidity in the elderly, the aging populations are at higher risk [45] of contracting COVID-19. Contrary to the claim [17] that susceptibility does not differ by age, furthermore, we found that the incidence rates relative to the population size of age groups manifest substantial variation (cf. Figure 7) as the epidemiology evolves, i.e., the susceptibility by age depends on specific signatures of mass infections and vaccination policies. Figure 7 shows that people over the age of 50 consistently had the most confirmed cases relative to their population size, until the vaccination prioritizing elderly population started in P5, thereby falling into the underrepresented group in the entire population. Minors (<20), conversely, consistently had the least confirmed cases relative to their population size due to education policies aligned to prevent outbreaks at schools such as extended vacation and remote learning, but gradually increased their shares as more and more schools had to resume their onsite learning. This is the only group that had more share of linked cases than unlinked ones (cf. Table 5), presumably due to the limited mobility of minors and the fact that their social contacts are concentrated on school-related activities. People in their 20s had bigger representations in P1 and P2 because they were the major attendees of Shincheonji church and the nightclubs in Seoul. As reported [18] in the USA, lower engagement in mitigation behaviors such as wearing masks, keeping distance, and washing hands in this age group might be a contributing factor. Their shares are high again in P5 and P6, but this overrepresentation was due to the sudden decrease in the share of elderlies that received prioritized vaccinations in those periods. Note that multiple strategies [46,47,48] can be formulated for an optimal vaccine allocation to meet specific goals such as the minimization of death rate or incidence rate. While Appendix A show people in their 20s had a higher rate of infection than other age groups at the Shincheonji incident, the heatmaps in Figure 8 do not reflect this tendency due to the missing information on transmission links. As reported in previous studies [14,19], infection rate was the highest within the same age groups, whereas older age groups tended to infect younger age groups more (e.g., 40s infecting minors in each period) than vice versa. We do not observe transmission from younger adults to older as in the southern USA [49]. Compared to P4, the proportion of the 60–79 age group decreased in P5, and in P6, the proportion of the 40–79 age group decreased, which is the result of prioritizing vaccination to the elderly population (cf. Appendix A). Once again, considering the high percentage of unlinked cases, this assessment is also subject to change when more information is available.

There are some critical issues that affect the evolution of the COVID-19 pandemic, and one of them is SARS-CoV-2 variants. We presented several SARS-CoV-2 variants observed in Korea from 30 January 2020 to 27 February 2022, as shown in Appendix A [35]. Note that the alpha and beta variants were dominant until 15 July 2021 (red) and were followed by two main variants, the delta (blue) and omicron (purple). The first five vertical lines divide the periods in the manuscript, and the last, 21 December 2021, indicates the time when the omicron exceeded 50%. It was discovered that the new delta variant D614G had higher infectivity in other countries [50] such that the transmissibility became twice faster in the delta variant than the alpha variant [51]. A recent study showed that transmissibility became faster in the omicron variant than the delta variant [52]. This is clearly shown in Appendix A: confirmed cases were dramatically increased after the omicron variant emerged (see January and February 2022). Furthermore, vaccine evasion has been reported for both the delta and omicron variants [6]. Therefore, this has important implications for the evolution of the pandemic and should be carefully endorsed by public health authorities.

Our work is not exempt from limitations. First, we have not carried out an extensive analysis of several SARS-CoV-2 variants. The evolutions of SARS-CoV-2 variants have been reported, and their molecular processes affect adaptation, transmissibility, host–pathogen generic characteristics [53]. Second, we have estimated $R_{t}$ using the built-in library in Python with only positive serial intervals. This $R_{t}$ could be different from the ones with negative serial intervals [52]. Lastly, there are some potential selection biases on data availability; for instance, only partial information of infection trees (infector–infectee) was used for our analysis, and the testing rate might be low at the beginning of the pandemic due to the shortage of testing resources.

## 5. Conclusions

We undertook a long-term study of COVID-19 infection in South Korea based on the data that are most up to date and currently available, and provided an in-depth analysis on the various infection characteristics such as the time-varying effective reproduction number, statistical information of unlinked and asymptomatic cases, infection networks, serial interval distributions, diagnostic delays and infection degrees, age-specific infector–infectee matrices, and infection profiles by age and region. We found that the case numbers had a positive correlation with unlinked cases but not with asymptomatic cases; social gatherings were a more common source of infection than family or work; serial intervals were shorter than other countries; diagnostic delays became shorter as the pandemic proceeded; and infection degrees were extremely positively skewed. We expect that our findings will help provide insights that are crucial to formulate age/region/time specific strategies that can cope with future pandemic development.

## Figures and Tables

**Figure 1 ijerph-19-04056-f001:**
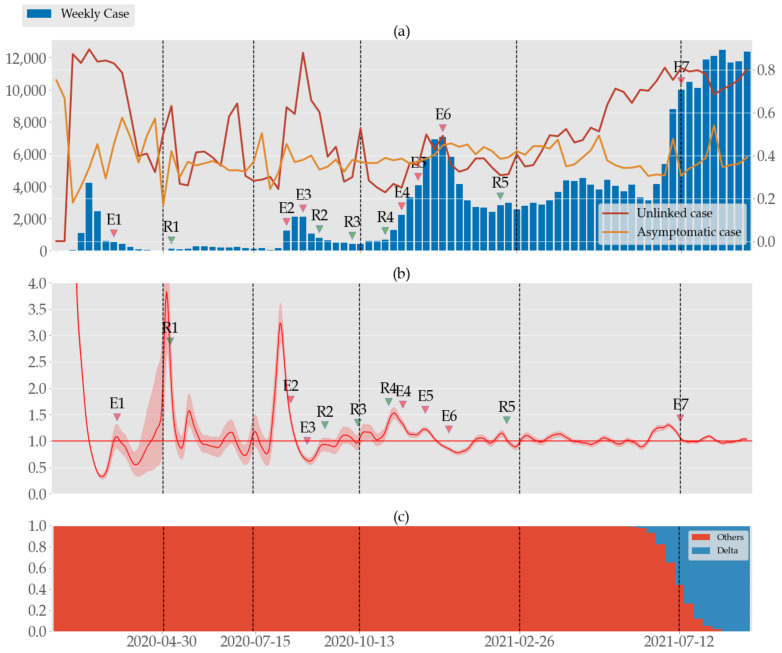
(**a**) Weekly number of local cases (blue bars) together with the proportions of unlinked cases (red) and asymptomatic cases (orange) in South Korea from 19 January 2020 to 16 September 2021, divided into six periods by vertical dashed lines. (**b**) Estimated daily effective reproduction number $R_{t}$ and its 95% confidence interval. The red horizontal line marks the critical threshold $R_{t}=1$. The red/green arrows mark the strengthening/relaxation of social distancing policies. See Appendix A for a detailed explanation of the policies. (**c**) Rates of the delta variant and the preceding ones such as alpha and beta [34,35].

**Figure 2 ijerph-19-04056-f002:**
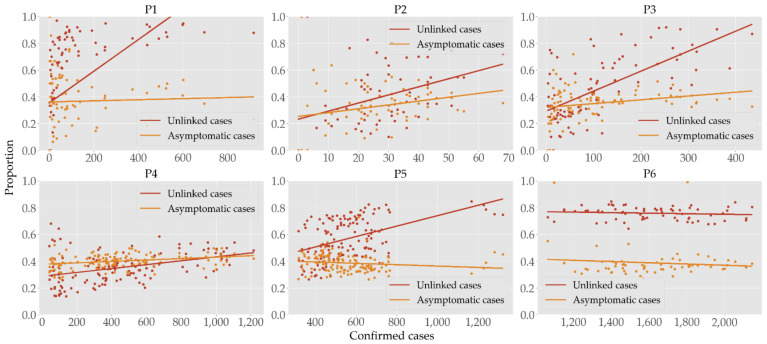
Scatter plots of daily confirmed cases versus the rates of unlinked (red) and asymptomatic (orange) cases. Visually comparing the slopes of regression lines across different periods can be misleading since the horizontal axes are not of the same scale.

**Figure 3 ijerph-19-04056-f003:**
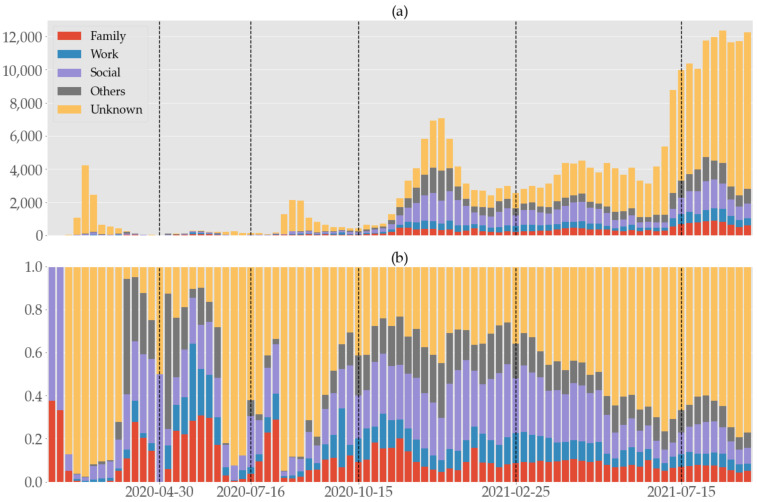
Weekly (**a**) numbers and (**b**) rates of confirmed COVID-19 cases categorized according to the infection relationship in South Korea from 19 January 2020 to 16 September 2021, divided into six periods by the vertical dashed lines.

**Figure 4 ijerph-19-04056-f004:**
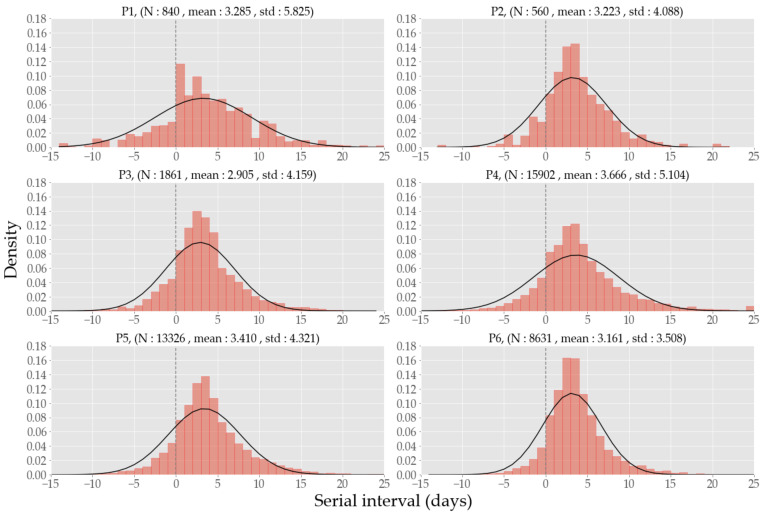
The serial interval distribution and its normal distribution fit in each period.

**Figure 5 ijerph-19-04056-f005:**
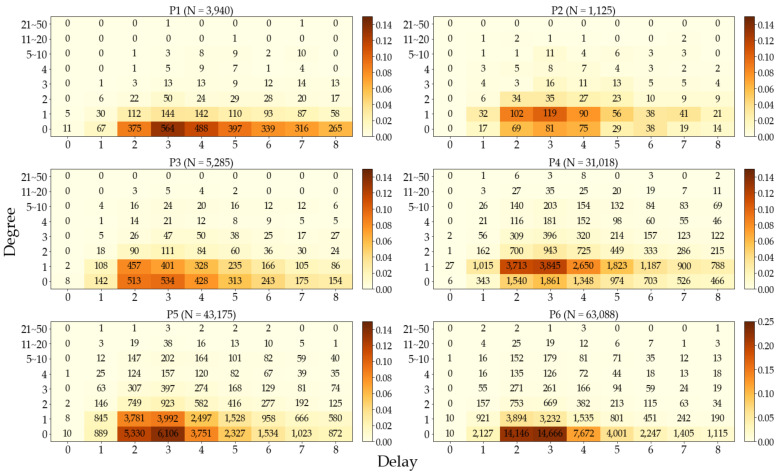
Cluster heatmaps for delay and degree where cases with delay >8 or degree >50 are truncated (23.7%, 9.2%, 10.5%, 10.4%, 6.5%, and 2.9% of total cases of each period). The density ranges from 0–0.14 for P1–5 and 0–0.25 for P6.

**Figure 6 ijerph-19-04056-f006:**
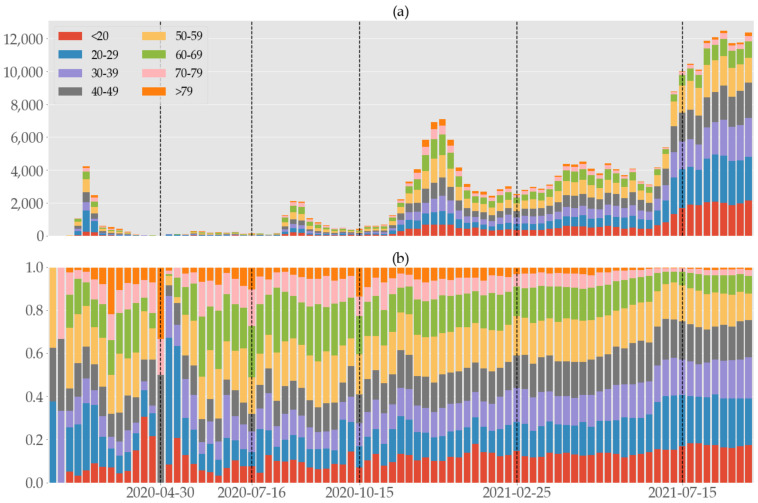
Weekly (**a**) numbers and (**b**) rates of confirmed COVID-19 cases segmented according to age brackets in South Korea from 19 January 2020 to 16 September 2021, divided into six periods by the vertical dashed lines.

**Figure 7 ijerph-19-04056-f007:**
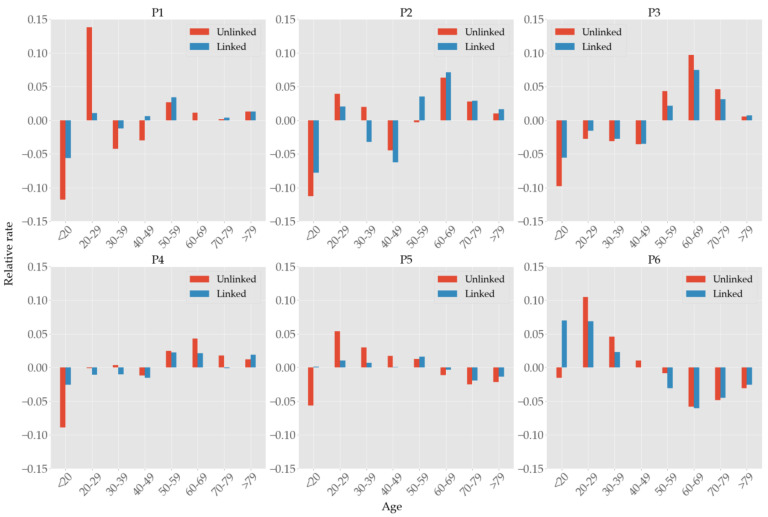
Shares of confirmed cases in each age group relative to its population size. For example, the unlinked cases of <20 in P1 is −11.8%, meaning that the difference of its infection share and population share is −0.118.

**Figure 8 ijerph-19-04056-f008:**
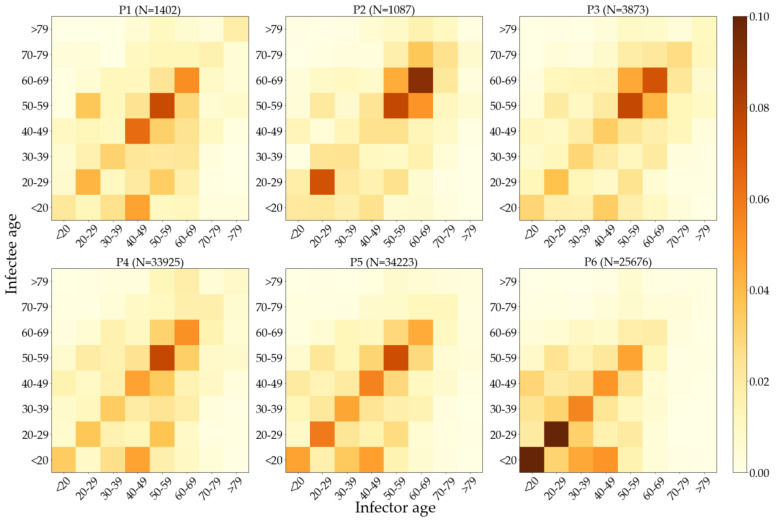
Heatmaps of the relative frequency of infector–infectee pairs with respect to their age.

**Figure 9 ijerph-19-04056-f009:**
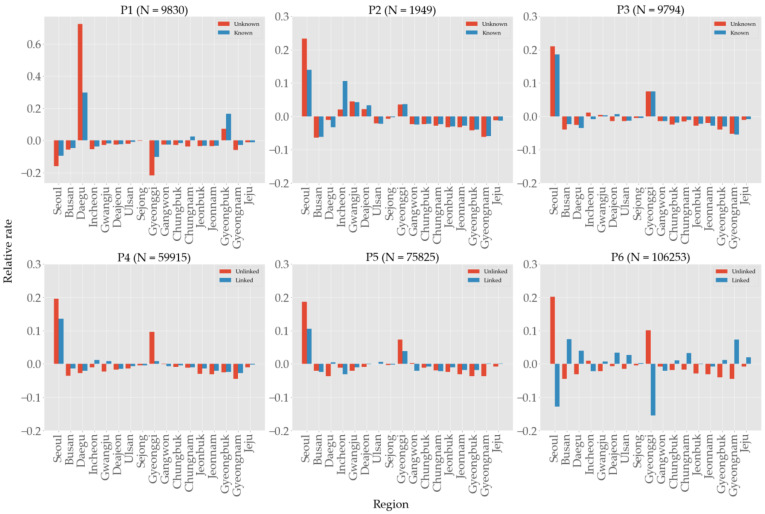
Regional shares of confirmed cases relative to population size. See Appendix A for confirmed cases per 1000 people.

**Table 1 ijerph-19-04056-t001:** Spearman correlation coefficients r for unlinked and asymptomatic cases vs. confirmed cases with *p* values (See Appendix A for the linear regression).

Period	Unlinked		Asymptomatic	
	r	*p*	r	*p*
1	0.779	0.002	−0.571	0.041
2	0.200	0.555	0.336	0.312
3	0.758	0.003	0.236	0.437
4	0.507	0.027	0.498	0.030
5	0.559	0.010	−0.211	0.373
6	−0.550	0.125	0.567	0.112

**Table 2 ijerph-19-04056-t002:** The percentage of confirmed cases according to infection relationship.

Type	P1	P2	P3	P4	P5	P6
Family	1.0	17.2	5.0	8.5	8.4	6.4
Work	0.9	15.0	4.7	9.2	7.8	5.2
Social	5.2	15.9	10.7	27.7	18.7	11.1
Others	2.7	13.3	3.6	20.7	11.0	9.3
Unknown	90.1	38.6	76.0	33.9	54.0	67.9

**Table 3 ijerph-19-04056-t003:** Statistics of diagnostic delays and infection degrees. Their medians and means appear to be higher than what the heatmaps of Figure 5 suggest due to the truncated cases with delay >8 or degree >50.

Period	Delay		Degree	
	Median	Mean	Median	Mean
1	6	6.58	0	0.40
2	4	4.77	1	1.26
3	4	4.88	1	0.93
4	4	5.00	1	1.38
5	3	4.28	0	0.84
6	3	3.51	0	0.41

**Table 4 ijerph-19-04056-t004:** Correlations between weekly number of diagnostic delays and infection degrees.

Period	Spearman		Pearson	
	Coefficient	*p* Value	Coefficient	*p* Value
1	−0.1513	0.6217	−0.1261	0.6815
2	−0.3909	0.2345	−0.6092	0.0466
3	0.1703	0.5780	0.1747	0.5681
4	−0.4195	0.0655	−0.4301	0.0584
5	0.7489	0.0001	0.7858	0.0000
6	−0.6606	0.0376	−0.7237	0.0180

**Table 5 ijerph-19-04056-t005:** Confirmed cases assorted according to age, gender, period, and link status.

	P1 (9830)		P2 (1955)		P3 (9834)		P4 (60,400)		P5 (76,021)		P6 (106,805)	
	Unlinked	Linked	Unlinked	Linked	Unlinked	Linked	Unlinked	Linked	Unlinked	Linked	Unlinked	Linked
	8289	1541	861	1094	5928	3906	22,937	37,463	44,784	31,237	80,600	25,903
<20	466	182	51	103	432	448	1825	5361	4951	5256	12,092	6095
	5.6%	11.8%	5.9%	9.4%	7.3%	11.5%	8.0%	14.3%	11.1%	16.8%	15.0%	23.5%
20–29	2230	219	147	166	613	451	2983	4504	8253	4408	18,926	5142
	26.9%	14.2%	17.1%	15.2%	10.3%	11.5%	13.0%	12.0%	18.4%	14.1%	23.5%	19.9%
30–39	770	190	133	112	607	413	3122	4585	7214	4316	14,207	3979
	9.3%	12.3%	15.4%	10.2%	10.2%	10.6%	13.6%	12.2%	16.1%	13.8%	17.6%	15.4%
40–49	1091	258	100	108	740	489	3399	5411	7910	4997	13,622	4104
	13.2%	16.7%	11.6%	9.9%	12.5%	12.5%	14.8%	14.4%	17.7%	16.0%	16.9%	15.8%
50–59	1605	310	141	221	1245	736	4398	7086	8016	5700	12,755	3522
	19.4%	20.1%	16.4%	20.2%	21.0%	18.8%	19.2%	18.9%	17.9%	18.2%	15.8%	13.6%
60–69	1120	191	163	216	1337	792	3999	5704	5536	4123	6363	1982
	13.5%	12.4%	18.9%	19.7%	22.6%	20.3%	17.4%	15.2%	12.4%	13.2%	7.9%	7.7%
70–79	593	114	85	109	694	399	2044	2651	2088	1631	1882	700
	7.2%	7.4%	9.9%	10.0%	11.7%	10.2%	8.9%	7.1%	4.7%	5.2%	2.3%	2.7%
>79	414	77	41	59	260	178	1167	2161	816	806	753	379
	5.0%	5.0%	4.8%	5.4%	4.4%	4.6%	5.1%	5.8%	1.8%	2.6%	0.9%	1.5%
Male	3191	711	447	517	2699	1842	11,567	18,209	23,399	15,351	43,333	13,783
	38.5%	46.1%	51.9%	47.3%	45.5%	47.2%	50.4%	48.6%	52.2%	49.1%	53.8%	53.2%
Female	5098	830	414	577	3229	2064	11,370	19,254	21,385	15,886	37,267	12,120
	61.5%	53.9%	48.1%	52.7%	54.5%	52.8%	49.6%	51.4%	47.8%	50.9%	46.2%	46.8%
Asymptomatic	3268	447	271	416	2082	1645	7684	17,094	15,314	13,736	27,452	13,487
	39.4%	29.0%	31.5%	38.0%	35.1%	42.1%	33.5%	45.6%	34.2%	44.0%	34.1%	52.1%
Delay (mean)	6.72	5.93	5.06	4.05	4.91	4.82	5.3	4.78	4.28	4.27	3.57	3.23

## Data Availability

The daily case data are publicly available from the Ministry of Health and Welfare, South Korea at http://ncov.mohw.go.kr (accessed on 18 February 2022) [54] and the Korea Disease Control and Prevention Agency (KDCA) [24].

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
