# Peer review of "Evolution of Responses to COVID-19 and Epidemiological Characteristics in South Korea"

_ijerph, 2022, doi:10.3390/ijerph19074056_

Round 1

Reviewer 1 Report

Overall, it is a useful study that provides insights on the roles of the different age groups in the various stages of the pandemic. The manuscript was understandable, although the language was not very clear at some points. I would recommend to the authors to have a native English speaker re-check the manuscript.

Major concerns/corrections:

Figure 1: instead of top bottom, the authors could use A, B. It would also be useful here to include 1C, to show the percentage of variants of concern. The authors can obtain this from NEXTSTRAIN (doi:10.1093/bioinformatics/bty407).

https://nextstrain.org/ncov/gisaid/global?f_country=South%20Korea

Table 1, Figure 2 and Table 2 are redundant. The authors should choose to show either the Spearman or Pearson table and put the rest in supplementary material.

Figure 3 would be more informative if below it there was the equivalent graph 3B with stacked bars up to 100% for each week.

Line 256-259: It is possible that the very high number of Unknowns in P1 could be related to low testing coverage.

It may be useful here to identify which period had the highest testing coverage (tests per cases) and for this one identify which categories played a significant role in infections.

Section 3.5. As the results are currently shown, it is not clear if there is a relationship between Diagnostic Delay and Degree. This part needs to be improved. Maybe the authors could do something else: Take as a data point (for each week) the average delay for each degree, and see if there is any correlation.

Figure 6 also needs a 6B with stacked bars up to 100%.

Line 311-313: The authors should do a statistical test or a pearson correlation (% vs week and get the p-value) to check this, but not leave it as it is unclear.

Figure 9 would be more understandable/digestible to the general reader if it showed unknown/known cases per million people or thousand people for each region and period.

It would be useful to add a section at the end of the Discussion that the different variants of concern have different infection rates and abilities to evade vaccination. This has important implications for the evolution of the pandemic. Given that SARS-CoV-2 is able to rapidly evolve (see doi: 10.3390/v14010078), as Omicron has recently shown, genomic screening should be endorsed by governments/policy makers.

I could not find the supplementary material.

Minor concerns/corrections

Line 18-20: It may be useful to also include variant content in these periods, from Nextstrain.

Line 31: “The biggest challenge” is a bit dramatic statement. At least, the authors should give a time frame (i.e. in the last X years).

Line 35: Maybe the authors could include these references for Delta and Omicron:

doi:10.1038/s41586-021-03944-y

doi:10.1016/S2213-2600(21)00559-2.

Line 37: It may be useful to include here the reference of Cameroni et al that shows increased immune evasion of Omicron.

doi: 10.1038/s41586-021-04386-2.

Line 41-43: please rephrase.

Line 46: This sentence is almost a repetition of a previous one.

Lines 46-51: Is this based on cases per 1 million people, or total numbers reported? I think a note should be inserted here that these numbers may be skewed by testing coverage in each country.

Line 54: as of 1st November 2021…

Line 56: I think it would be useful here to include again the Cameroni reference concerning Omicron’s antigenic shift and another reference on the high infectivity of Omicron (see doi:10.1016/S2213-2600(21)00559-2.).

Line 157-160: please rephrase.

Line 181: eclipse 4,000?

Line 191: I think it would be useful to mention here that the infection rate of Delta was significantly higher than the original Wuhan-Hu-1 strain, with an R0 around 5-7 (see doi:10.1016/S2213-2600(21)00559-2.).

It may also be useful to mention somewhere that around March/April 2020 emerged the D614G mutation in spike that increased the infectivity (doi:10.1016/j.cell.2020.06.043). That was the first mutation that was related to significantly increased infectivity.

Line 199-201: please rephrase.

Lines 246-248: Could you please clarify if School is also within the “Others” group?

Line 273: It would be useful here to report what numbers did other similar studies found.

Figure 8 is very informative and interesting. It shows that most infections occur within the same age group, with the exception of the younger group.

Reviewer 2 Report

Using the data gathered by the Korean government, the manuscript aimed to understand the impact of different major events, including interventions, on a number of epidemiologic metrics. The work falls short in three aspects; (i) methodologies used are too simplistic hence findings are not robust/reliable; (ii) the format and structure of the manuscript needs to be significantly improved. The Materials and Methods section aims to provide the audience the definitions of measures and the statistical methods used to produce the results. The Results section solely aims to objectively describe to results produced by the methods described. Any information on how data are used, how measures are constructed; and how measures are computed should be discussed in the Materials and Methods section. (iii) reliability of the data and measurements: asymptomatic patients are less likely to get tested so there is selection bias in the study. It is suggested to only focus on symptomatic patients. The fact the asymptomatic got tested was possibly due to the linkage with a confirmed case/cluster, creating a spurious association between asymptomatic and linked cases.

I personally found Figure 9 very interesting and very useful. It should help understanding the transmission of SARS-CoV-2 and public health policy making.

Please also find below specific comments.

Section 2. Materials and Methods: there is a serious lack of how data were used and what measures were used and computed. While some metrics and measures were defined and introduced, I did not realise the outcome measure and the associated statistical methods used until I read the Result section. This section should describe the statistical methods used in detail. In particular, the estimation of Rt is too simplistic. The authors should consult an epidemiologist concerning the usage, pros, and cons of Rt.

Line 110-115: If a case is linked to a known cluster, it should be considered as “linked”. An identified infector is not a must; an identified cluster is sufficient. See PMID: 34591778

Line 124-127: This should be addressed in the Materials and Methods section.

Table S1: How were the 6 periods related to the events listed? Please clarify.

Line 157-160: The statement on poor ventilation is not proved by the data. It is highly speculative and should rather be addressed in the Discussion section.

Line 167: by how much in % the cases of the church accounted for?

Line 217-222: this comparison is too simplistic. Also in P1 of Figure 2, the variance is higher in the lower end hence the slope is misleading (heteroskedasticity). Then in P5, there are clusters of outliers.

Table 2: Please provide the p-value of the slope coefficients.

Line 268-273: How was the normal distribution fitted to the data? This should be addressed in the Materials and Methods section. Also, please provide the p-value of normality test and describe the type of normality test used.

Discussion section: Some limitations should be addressed, such as potential selection bias. Moreover, testing rate might be low in the beginning of the pandemic owing to the shortage of testing resources.

Round 2

Reviewer 1 Report

The authors have addressed my concerns and also performed all the suggested corrections.

Reviewer 2 Report

I have no further comments.